# Quo Vadis Smart Charging? A Literature Review and Expert Survey on Technical Potentials and User Acceptance of Smart Charging Systems

**Julian Huber** [1,2,*], **Elisabeth Schaule** [1,2], **Dominik Jung** [3] **and Christof Weinhardt** [1,2]

1   FZI Forschungszentrum Informatik, 76131 Karlsruhe, Germany; elisabeth.schaule@gmail.com (E.S.);
    christof.weinhard@kit.edu (C.W.)
2   Karlsruhe Institute of Technology, 76131 Karlsruhe, Germany
3   Porsche AG, 74321 Bietigheim-Bissingen, Germany; d.jung@kit.edu
*   Correspondence: julian.huber@fzi.de

**Abstract:** Uncontrolled charging of plug-in Battery Electric Vehicles (BEV) represents a challenge for the energy system. As a solution, recent studies propose smart charging to avoid grid congestion and to integrate renewable energy. While financial benefits for smart charging schemes are currently quite low, there are other objectives for smart charging. However, it is unclear for which objectives smart charging can be used most effectively and which arguments are most likely to convince end users of BEVs to use smart charging schemes. To fill this gap, we conducted a literature review of the premises and the objectives of smart charging and how they fit the end-user's motivation to use such smart charging systems. To evaluate the results, we present findings of 16 domain experts who evaluated various statements on smart charging according to their technical correctness and their persuasiveness towards end users. The results show that experts consider those smart charging objectives as most persuasive towards end users which they consider technically correct. Moreover, cost savings and integration of renewable energies are rated highest on both scales. On the contrary, experts do not expect a positive impact of smart charging systems on battery life and rate it as not very convincing.

**Keywords:** electric vehicles; smart charging; objectives; end-user

---

## 1. Introduction

Road transportation accounts for 20% of global $CO_2$ emissions and contributes to the emission of air pollutants within cities. Battery Electric Vehicles (BEV) do not emit any local air pollutants and can be operated as $CO_2$ neutral. Therefore, governments and industry have set ambitious goals for the diffusion of BEVs [1]. However, compared to other residential electric consumers, BEVs have a high peak demand. Moreover, many BEV users charge their BEVs in the evening resulting in a high degree of simultaneity of the charging demand. These peaks can challenge the electric grid even at low penetration rates [2].

Recent research argues that there is flexibility within the charging process that could be used to flatten load peaks and to achieve other optimization objectives. While a grid operator can control charging to avoid grid congestion and thus expensive network expansion, an electricity supplier can use flexibility to react in price spikes or lower balancing costs. Furthermore, they can shift the charging process towards times with lower shares of conventional generation within a generation portfolio, which can result in lower prices and $CO_2$ emissions during charging [3].

However, smart charging requires the BEV users to accept a certain amount of flexibility in their charging process. The acceptance of flexibility in the charging depends on the objectives implemented

in the smart charging system [4]. Therefore, the *EVS32* paper [5] provides a literature review and expert survey on objectives of smart charging systems. In this paper, we expand the results presented in [5] by adding an analysis of the technology and flexibility that influences the design of smart charging systems.

In the following, we define a smart charging system as an information system that optimizes the charging process towards one or multiple objectives besides reaching a desired state of charge (SoC) within a given time frame. Typical objectives are technical, financial, and socio-environmental goals [6]. Both the technical properties of the charging system (i.e., the BEV and the charging station) and the willingness of end-user to accept a delay in charging or a reduction in the final SoC restrict the solution space for the optimization of these objectives. While the technical restraints are fixed, the BEV driver can either input this restrictions explicitly in some of the smart charging systems (e.g., via a smartphone application) or accept a certain degree of smart charging by choice of her charging tariff (e.g., Ensslen et al. [7]).

However, the acceptance for the use of such systems depends on the objective of the smart charging system. For instance, Will and Schuller [8] report that smart charging systems are more likely to be accepted if they ensure grid stability and integration of renewable energy resources (RES) instead of solely financial benefit. Based on the success of the energy-conservation programs, Huber et al. [9] propose that message framing could nudge people to become more flexible using a smart charging system. The findings of these studies provide evidence that the choice of the objective function can be an incentive for end users to use smart charging systems and charge with higher flexibility.

Therefore, successful business models based on smart charging systems must draw together the end users' motivation to use such systems and a rewarding objective function for the operator of the smart charging system. The number of possible objectives opens an ample solution space for the design of smart charging systems. To provide guidance and give design recommendations for practitioners and scholars, in this paper, we investigate the most promising applications for smart charging based on a literature review and a survey of domain experts. For that purpose, this study answers the following research questions:

*RQ1: What are the objectives of charging system operators present in academic literature?*

*RQ2: What are promising incentive factors to motivate BEV drivers to use smart charging systems?*

*RQ3: Do the most promising objectives of smart charging system operators fit the BEV driver's motivation to use smart charging systems?*

The remainder of the paper is structured as follows: The paper opens with a description of the technology and flexibility in smart charging in Section 2. In Section 3, we conduct a structured literature review based on the methodology of Webster and Watson [10] to answer *RQ1* and *RQ2*. We investigate the abstracts of 1.056 articles from the domains of computer science, economics, and engineering between 1980 until today.

In a second step, in Section 4, we conduct an expert survey to validate the literature review and answer *RQ3*. Based on eight incentive factors for smart charging found in the literature review, we derived statements on the benefits of smart charging. We asked 16 domain experts to evaluate these statements on their technical correctness and their persuasiveness for end users. In the last step, in Section 5, we summarize the results and provide recommendations for information system researchers and designers of smart charging systems.

## 2. Status Quo

The flexibility in smart charging comes from the technical aspects of the BEV and the charging station and the behavior of the BEV driver. This section provides a review of the

technology and flexibility used in smart charging to enhance the understanding of the technical and behavioral constraints.

### 2.1. Technology in Battery Electric Vehicles

The BEVs' type and battery technology provide the first technical restraint to smart charging. This subsection provides a summary of contemporary BEVs and their battery technology and their suitability for smart charging.

#### 2.1.1. Types of Battery Electric Vehicles

Electric cars are automobiles for which electric motors provide the propulsion energy to the wheels. Subgroups are hybrids or hybrid electric vehicles that have two storage systems for propulsion energy. First, a gas tank feeding an Internal Combustion Engine (ICE) or fuel cell and an electric rechargeable battery unit powering the electric motors. While pure hybrids only recharge during driving, e.g., by recuperating braking energy, plug-in hybrids can also recharge the battery system connecting it to the electric grid and are therefore interesting for smart charging. Full-electric vehicles (FEVs) do not have a secondary storage and conversion unit beside the rechargeable battery unit. Table 1 shows the technical specifications of most-selling BEVs in the US in 2019.

As only PHEVs and FEVs charge form the electricity grid, only they can have an impact on the electricity system, e.g., by causing load peaks or providing flexibility by demand-side management measures. In consequence, the following work focuses on PHEVs and FEVs, which are both summarized by the term BEV.

Because they do not have a conversion unit, e.g., ICE or fuel cell, FEVs do not emit any local emissions, except noise and heat. The same applies to hybrid vehicles operated in full-electric mode. However, as hybrids usually have much smaller battery units, they usually only have an electric range of up to 50 km (see Table 1) and are therefore less interesting for smart charging.

**Table 1.** Specifications of best-selling BEV in the US (in 2019).

| Model | Battery Capacity [kWh] | Efficiency [kWh/100 km] | Price [$] | Type | Source |
|---|---|---|---|---|---|
| Tesla Model 3 RWD | 50.0 | 15.6 | 38,990 | FEV | [11] |
| Toyota Prius Prime LE | 9.0 | 15.8 | 27,600 | FEV | [12] |
| Tesla Model X AWD 90D | 90.0 | 20.7 | 84,990 | FEV | [13] |
| Chevrolet Bolt EV | 60.0 | 17.6 | 36,620 | FEV | [14] |
| Tesla Model S AWD 100D | 100.0 | 20.6 | 70,115 | FEV | [15] |
| Honda Clarity PHEV | 17.0 | 19.0 | 33,400 | PHEV | [16] |
| Nissan LEAF (40 kWh) | 40.0 | 18.7 | 29,990 | FEV | [17] |
| Ford Fusion Energi | 7.0 | 20.5 | 34,595 | PHEV | [18] |
| Chevrolet Volt | 18.4 | 19.5 | 33,520 | PHEV | [19] |
| BMW 530e | 9.2 | 28.5 | 53,900 | PHEV | [20] |

#### 2.1.2. Battery Technologies

The battery units in BEVs apply different battery technologies. These concepts differ in several properties. Besides cost, energy density efficiency is one of the most discussed properties. Energy density efficiency qualifies how much electric energy a battery unit can store at a given volumetric size or weight. For instance, gasohol E10, i.e., gasoline with 10-volume-% added ethanol, has a specific energy of 12,094.5 Wh/kg and an energy density of 9,216.7 Wh/L. In contrast, a lithium-ion battery ranges around 100.00–243.06 Wh/kg and 250.00–730.56 Wh/L.

As batteries have a low energy density FEVs require larger and heavier energy storage systems than cars with an ICE. However, this is partly compensated be the higher tank-to-wheel efficiency of FEVs. While ICE efficiency is limited by the temperature difference of Carnot Efficiency and max out at 25–35%, electric power trains in FEVs can exceed 90% tank-to-wheel efficiency [21].

As a BEV accelerates and slows down perpetually, a lower weight of the vehicle improves, besides others, energy efficiency, driving dynamics, and wear. Therefore, one key factor in the selection of battery technologies is energy density efficiency in both weight and volume.

From the point of smart charging the optimal battery, has specific characteristics. The battery should have little self-discharging so that the battery can be charged to full SoC well before the departure, e.g., when energy is available, and hold this energy until the time of departure. Next, the battery should have no memory effect, i.e., a reduction in capacity if the battery does not fully discharge before starting the next charging cycle. Using a battery with memory effect would imply that (smart) charging should be only conducted at a low SoC to retain battery life, reducing the potential for smart charging. Especially with vehicles-to-grid concepts, low cyclic ageing is an essential factor. Otherwise, using the BEVs battery as buffer storage would directly reduce the batteries life expectancy.

Manzetti and Mariasiu [22] describe the characteristics of various battery technologies in the order of increasing energy density. Lead-acid (Pb-acid) battery systems are the oldest electric energy storage technology used in car, as they are commonly used as starter batteries in cars with ICE. While being inexpensive, a downside is the usage of acid substances within the car. Compared to other technologies, Nickel-Cadmium (NiCd) batteries have the upside of showing low cyclic ageing, which is a benefit in smart charging, especially vehicle-to-grid (V2G) concepts. The characteristics of Nickel-Metal-Hydride (NiMH) batteries resemble Nickel-Cadmium. However, in comparison, they show a lower memory effect. A characteristic negative to the use of smart charging is that such batteries show a high amount of self-discharging. High self-discharging implies that the battery should be charged right before the departure of the BEV so that the driver can profit from a full SoC. While Sodium Nickel Chloride (NaNiCl) batteries can store electricity for more prolonged periods, they are linked to problems with operational safety. Lithium-ion polymer batteries show lower cyclic ageing than standard Li-ion batteries. While they are well suited for both FEV and smart charging because of low cyclic ageing and no memory effect, technical challenges are functional instability against overloading and deep discharging.

### 2.1.3. Battery Ageing in Lithium-Ion Batteries

Lithium-ion polymer is the most popular battery technology in BEVs, e.g., all BEVs in Table 1, are applied with lithium-ion battery technology. Given the characteristics of such lithium-ion batteries, they are well suited for storing propulsion energy in cars due to their high energy density. They also allow an interrupted charging and starting charging processes different SoC levels which is favorable for smart charging. Still, the operation and charging behavior of BEVs impacts the expected lifetime of lithium-ion batteries (see e.g., Vetter et al. [23], who provide a detailed description of the chemical processes leading to battery ageing).

Barré et al. [24] differentiate between calendar and cyclic ageing. Calendar ageing describes the loss in energy capacity that is independent of the cycling of the battery, i.e., how often the battery charges and discharges. The main factor how fast a batteries energy capacity degrades with lifetime is the batteries temperature. In general, lower battery temperatures result in slower calendar ageing. This ageing process is independent of the way the battery operates during smart charging.

In contrast, cycle ageing describes all factors concerned with the use of the battery and is therefore influenced by smart charging. In particular, the upper and lower limits in the energy capacity used for cycling and the charging current influence the cycle ageing of lithium-ion batteries [25]. Using a more extensive range of SoC and higher voltage levels to increase maximum charging power in smart charging increases the flexibility but increases cycle ageing. Therefore, there are trade-offs between the usage of BEVs as demand-side management measures to provide services to the energy system and the battery lifetime of BEVs.

Battery ageing is relevant, as current discussions involve the ecological life-cycle assessment of FEVs compared to cars with ICE. Critics claim the resource-intensive production of FEV batteries and the usage of non-renewable electricity for charging FEVs has adverse effects of the increasing number

of FEVs. The results of the life-cycle assessment highly depend on the expected lifetime of the FEV. Hawkins et al. [26] calculate that starting from lifetimes of 150.000 km, the climate impacts of PHEVs and FEVs are 27% and 78% lower compared to cars with ICE.

*2.2. Smart Charging*

BEVs rely on the electricity system to meet their energy requirements. As maximum charging power of BEVs is high compared to other domestic energy consumers, regulators and industry derived particular plug types (defined in IEC 62196) and communication protocols to ensure safe handling of high voltage and currents occurring with BEV charging. Conductive charging systems are the most common charging system in which a cable-plug connection transmits the electricity. Meanwhile, inductive charging, i.e., wireless charging, uses electromagnetic induction to transfer energy between induction coils.

2.2.1. Charging Technology and Standardization

IEC 61851-1 defines four different charging modes for conductive charging systems with different voltage level, thus maximum charging power and communication capability. While Modes 1–3 operate with alternating current, Mode 4 provides direct current to the BEV. Table 2, adapted from Hardman et al. [27], provides a review of charging modes defined in IEC 61851-1.

As a default, Mode 1, describes the charging of BEV on a local household socket outlet or a single or three-phase CEE-socket. By treating, the BEV as a straight electric consumer, this offers a simple fallback solution as a BEV with charging Mode 1 can charge at any household socket. On the downside, this charging mode is rather slow, and no protocol for charging coordination is applied. Therefore, this Mode 1 is not suited for smart charging. Mode 2 relies on a dedicated delivery point (socket outlet), i.e., wall box, procuring higher maximum charging power and allowing for communication between delivery point and BEV. While Mode 2 delivery points are often equipped in residential and work areas, the faster Mode 3 delivery points are mostly found in workplaces and public charging locations. The reason is that their maximum charging power exceeds the power capacity of typical residential house electricity connections. The communication protocol in Mode 3 bases on IEC 61851-1 or ISO/IEC 15118. Delivery points with Mode 4 charging provide direct current at high voltage levels up to 400 kW.

**Table 2.** Charging Modes defined in defined in IEC 61851-1 adapted from Hardman et al. [27].

| Charging Mode | Power [kW] | Smart Charging | Typical Location | Socket System [Outlet\|Inlet] |
|---|---|---|---|---|
| Mode 1 | 1–3 | No | Home | Domestic plug\|Type 1/2 |
| Mode 2 | 1–7 | Yes | Home, Work | Domestic plug\|Type 1/2 |
| Mode 3 | >43.5 | Yes | Work, Public | Type 1/2\|Type 1/2 |
| Mode 4 | >400 | Yes | Corridor | CCS (CHAdeMO) |

2.2.2. Flexibility in BEV Charging

BEV charging does often not require a predefined load profile but has some intrinsic flexibility. Flexibility can be *'defined by the ability to follow different paths of action at a given point in time to provide a service for another entity'* [28]. Daina et al. [29] use a similar Figure 1 to describe and model the flexibility in BEV charging based on charging choices of BEVs users. If the BEV user decides for a minimum $SoC_d$ at a given deadline $t_d$, the charging process of BEVs is flexible, as the energy demand can be fulfilled using different paths within the dotted area, stating at the time of arrival $t_a$. A smart charging system can realize different paths by influencing the charging power or the energy provided to or extracted from the battery.

Using the taxonomy provided by Petersen et al. [30], the charging process of a BEV can be modelled as a battery, where a certain state of charge $SoC_d$ must be reached within a time deadline, usually departure time $t_d$. The constraints of the charging process, presented in Figure 1, are the

maximum charging power $C^{max}$, the energy demand $SoC_d - SoC_a$, and the time available for charging $t_d - t_a$. Within these constraints (dotted area), the charging process (black line) can be optimized by smart charging systems. Vehicle-to-Grid concepts allow the discharging of the BEVs' batteries by allowing for negative charging powers, i.e., a dropping black line.

Neupane et al. [31] and Ludwig et al. [32] use the terms time and energy flexibility to describe the flexibility in energy consumption. Energy flexibility is the potential for change in the energy consumption profile, while time flexibility is the potential for a shift of the consumption profile.

Similarly, we define time flexibility in BEV charging as the time interval of reaching the desired $SoC_d$ at maximum charging power $C^{max}$ compared to the planned time of departure $t_d$. A simple metric for energy flexibility is the possible slack in total energy procurement during charging until the time of departure. Hence, there is no energy flexibility if a full battery ($SoC_f$) is required.

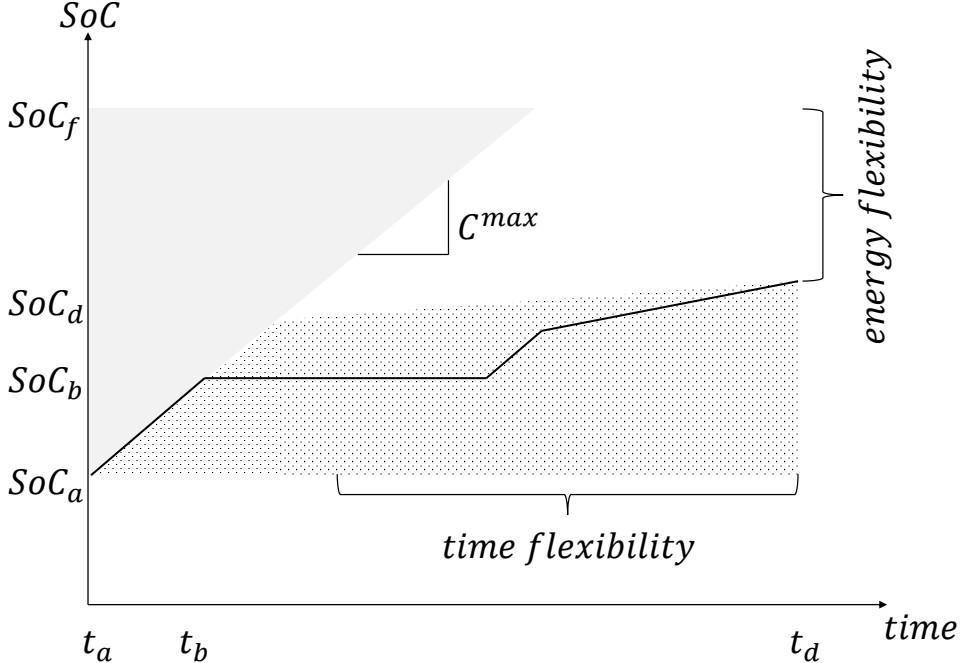

**Figure 1.** Schema for charging flexibility.

2.2.3. Flexibility and Mobility Requirements

Both energy and time flexibility in BEV charging depend on the BEV driver's mobility requirements. Long parking durations offer high time flexibility, while short trips rarely require full SoC and thus result in high energy flexibility.

Lunz and Sauer [33] analyze the driving behavior of German car users based on the trip diaries obtained by Zumkeller et al. [34]. Drivers in this panel drive on average 36 km (36.3 km in our evaluation of the same data set) per day. Almost all, i.e., 95%, single trips are shorter than 42 km, for the same share, the total trip distance is below 150 km. Consequently, Lunz and Sauer [33] argue that if BEV drivers charge their BEVs overnight, a BEV with 150 km range could cover 95% of trips of German residential car users. Adding fast recharging opportunities, i.e., >22 kW, after each trip, such a BEV could cover 99% of total trips. This analysis shows that even with smaller battery capacities, there is flexibility potential in the charging process, as trips are rather short compared to the parking duration of BEVs.

Quirós-Tortós et al. [35] conduct a similar analysis of BEV charging behavior of 221 residential BEV users in the UK. They report that users charge at full available charging power most of the time. Using the full capacity implies that no smart charging in applied yet as load shifting or curtailing would result in lower charging rates in some hours. On most days, i.e., 70% of days, BEV users connect

their BEV only once. While the first connection to the charger mostly happens at medium levels of SoC (between 25% and 75%), only 65% of fist connections end with a fully charged battery. In contrast, second connections result in full SoC more often.

Most drivers charge their BEV at home. A position paper of VDA [36] states that currently 85% charging processes are at private installation locations, i.e., residential and company parking, while only 15% happen in publicly accessible locations. However, this share is likely to rise to 30–40% within the next years.

Private parking locations are also the locations with the most extended parking duration. Own evaluations of the German mobility panel [34] show that parking locations of cars show a pattern though-out the day. On the weekday mornings, workplaces are the most likely parking location. At workplaces, cars on average stay unmoved for 6.2 h. Arrivals at home peak in the early evening hours. Parking durations at home are on average 13.9 h long. Also, they show a higher variance than parking durations at work. In contrast, parking durations at publicly accessible locations are rather low, e.g., 2.5 h for parking at shopping locations. Therefore, the highest time flexibility and potential for smart charging is found at the homes of BEV drivers. The average trip distance for car trips in Germany is 36.3 km. At an efficiency of 20 kWh/100 km (compare Table 1), the average residential car usage in Germany would require 7.4 kWh of electric energy. Most cars in Table 1 show higher battery capacity letting expect energy flexibility in BEV charging.

In summary, the current technology (lithium-ion batteries and Mode 2 charging) enables smart charging. A comparison of common battery sizes with the user behavior of car users also shows considerable potential for time and energy flexibility.

## 3. Literature Review

Overall, there is a vast amount of literature on *'Smart Charging'*. For example, Google Scholar yields 823.000 hits as of January 2019. In this section, we first describe related works and try to classify literature on smart charging. First, we acknowledge related reviews on smart charging. We then focus on *RQ1* and discuss recent objectives on smart charging systems. Last, we describe the results of the review of incentive factors of smart charging answering *RQ2*.

### 3.1. Reviews on Smart Charging Systems

Yilmaz and Krein [37] outline the technical environment by reviewing battery charger topologies, charging power levels, and charging infrastructure. The authors distinguish between unidirectional and bidirectional, i.e., V2G, energy and information flow. Unidirectional systems can provide reactive power and provide frequency regulation in one direction by curtailing the charging load. Bidirectional systems allow for additional use cases (e.g., replacement of spinning reserves). On the downside, such applications can age the battery faster due to the frequent circling.

Similarly, García-Villalobos et al. [38] provide a review of smart charging approaches where they differentiate between V2G and non-V2G concepts. They further analyze the main objectives of charging systems, and their solvers and tools, software, and strategy used. The paper lists frequency regulation, voltage regulation, generation costs, charging cost, provision of ancillary services, minimization of power loss, renewable integration, the adaption of load factor and variance, and optimized operation of distribution networks as possible objectives. Meanwhile, Mwasilu et al. [39] focus on the single objective of renewable energy sources integration and provide a review of the potential, impacts, and limitations of V2G technologies. Despite this broad overview, these reviews do not evaluate the frequency and BEV drivers' acceptance of smart charging objectives.

### 3.2. Operators Objectives of Smart Charging Systems

To answer *RQ1* we structure our literature review as follows. First, we select three literature databases covering the fields of computer science, economics, and engineering. As a search term we start with using *'vehicle ∧ charging'*. This search results in several thousand matches spanning

topics as charging protocols, route planning, and battery management systems. To narrow down the search space towards the objectives of smart charging systems, we add the terms *'objective* $\land$ *incentive* $\land$ *acceptance'*. The second search results in the initial 1.056 results of matches noted in Table 3.

**Table 3.** Matches for the search term in different data bases.

| Literature Databases | Search Term Vehicle $\land$ Charging $\land$ | | |
|---|---|---|---|
| | *Objective* | *Incentive* | *Acceptance* |
| ACM Digital Library | 17 | 6 | 10 |
| IEEE Explore | 422 | 75 | 69 |
| ScienceDirect | 319 | 120 | 98 |

We focus on a concept-centric approach, as proposed in Webster and Watson [10]. Using a subset of the initial 1.056 results of papers (the 422 matches from IEEE Explore on smart charging objectives), we analyzed their titles and abstracts to identify key concepts. As expected, most papers in the initial 1.056 results describe the design, optimization, and scheduling on the charging process of electric vehicles. However, one main difference is the role and perspective of the charging system operator, who is in charge of the smart charging system.

A fist group of papers focuses on the *grid-centered* perspective of grid or system operators who centrally control the charging process of many BEVs to provide system services, optimize power flow, dispatch, or avoid congestion in their grid (e.g., in Mojdehi and Ghosh [40]). Depending on the regulatory framework assumed in the paper, this can be either an integrated system operator who manages both energy generation and grid operation or grid operators solely optimizing grip operation.

The next group focuses on *market-cent red* perspective of charging system operators or aggregators, who coordinate the charging process of multiple BEVs to optimize their outcome at market level (e.g., matching charging with a product or generation portfolio or using the flexibility of the charging portfolio on reserve markets [41]).

Last, the third group of papers is *locally centered* and aims at an operator optimizing the charging of BEVs to match consumption with a local energy resource (e.g., in Mou et al. [42]). In this case, the BEV driver often is the same entity who operates the charging station (e.g., in a residential setting where the driver integrates the BEV into the home energy management system.

The smart charging systems pursue different objective functions according to the charging operator's perspective. We use the dimensions from Sovacool et al. [6] for a broader categorization:

Financial objective functions mainly result from a *market-centered* perspective and focus on cost advantages realized by optimized energy procurement considering benefits from the provision of auxiliary services and the pricing of charging services. The charging operator can achieve financial advantages by optimizing charging in line with changing prices on the energy markets (e.g., Limmer and Dietrich [43], Li et al. [44]). Additionally, some authors also consider using the flexibility in smart charging on frequency reserve markets [45]. One objective not mentioned by García-Villalobos et al. [38] but by Sovacool et al. [6] is the minimization of battery degradation. As battery degradation depends on the charging strategy, some authors propose to control charging in a battery protecting manner [46]. Others consider battery degradation a constraint to be considered in economic optimization [47]. Like Sovacool et al. [6] we consider battery degradation to be part of the financial dimension, since the battery life has a direct financial impact on the BEV owner and, unlike the other points of the technical dimension, is not related to the *grid-centered* perspective.

The technical objective functions arise from a *grid-centered* perspective. The technical dimension affects the financial dimension, if, as in Deilami et al. [48], an integrated system operator manages the generation dispatch and the power grid at the same time. In this case, the charging operator can obtain financial benefits from integrating BEV charging in grid operations. Besides matching the generation, smart charging can also be a tool in congestion management [42] and provide system stability in the form

of different ancillary services such as frequency regulation, voltage regulation, and minimization of power loss (see García-Villalobos et al. [38], Mojdehi and Ghosh [40], Mathur et al. [49], Staudt et al. [50]).

Likewise, flexibility in smart charging systems can also mitigate the uncertainty in wind [51] or Photo Voltaic (PV) [52] generation to integrate a higher share of RES and therefore can help to reduce carbon emissions of the energy generation [3]. Other authors consider fairness [43] in the scheduling of the charging loads and discuss community-based charging stations [53]. Socio-environmental objectives often play a role in works with *locally centered* perspective, e.g., when focusing on the integration of local renewable energies [54].

Figure 2 shows the main objectives of smart charging systems from the perspective of the operator of the smart charging system sorted by technical, financial, and socio-environmental focus. From the perspective of BEV drivers, the main objective is the fulfilment of their mobility needs. For the charging system operator, the fulfilment of the drivers' mobility needs sets the constraints for the optimization of smart charging systems.

Please note that some other structures and concepts can distinguish smart charging systems: The coordination can occur by decentralized or centralized control. Models differ in their assumptions on the flexibility of the charging process and model them either as an interrupting load or as batteries in V2G concepts. The scheduling of loads is based on different methods, such as heuristics, genetic algorithms, or optimization. However, those concepts do not help in answering the research question and are not further considered.

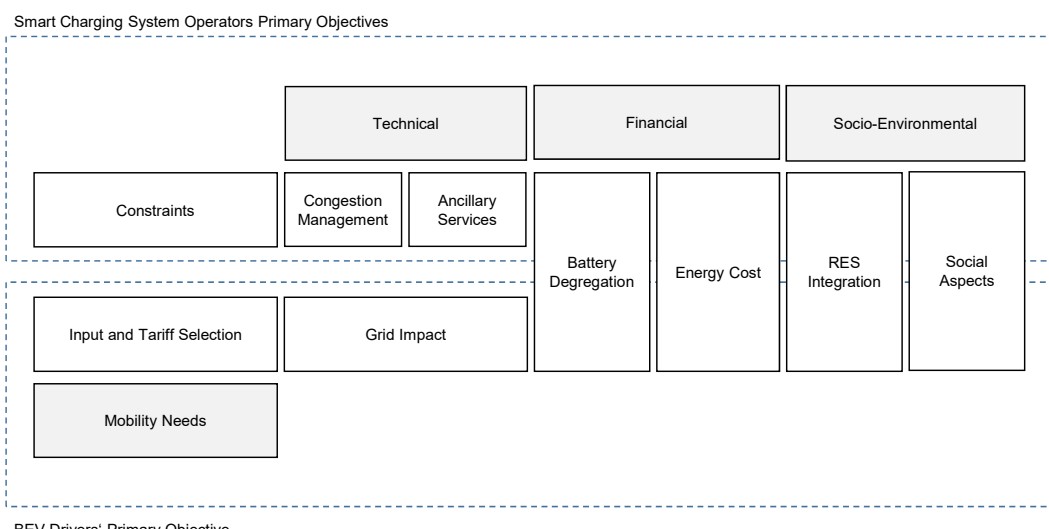

**Figure 2.** Objectives of Smart Charging System Operators and BEV Drivers.

### 3.3. Trend Analysis for Smart Charging Objectives

To identify prevailing smart charging objectives (*RQ1*), we next conduct a keyword search in the abstracts of the papers found in Table 3. Therefore, we analyze the occurrence of the keywords in the pre-processed, i.e., removing punctuation, lower casing and stemming, abstracts of the initial 1.056 literature matches. First, we generate a list of indicator keywords by screening the most common words, i.e., occurring more than 40 times, in the combined abstracts and assigning them, if relevant, to one of the optimization objectives. Table 4 lists the resulting keywords for different charging objectives.

We assume that the occurrence of a keyword indicates, whether the paper considers a given objective, e.g., a paper containing the words *'lifetime'*, *'life-time'*, *degradation'*, or *'aging'* is likely to consider battery degradation within the smart charging system. We conduct an automated search through the titles and abstracts in the initial 1.056 results. We discard all articles not containing any of the keywords as irrelevant.

**Table 4.** Objectives for smart charging.

| Objectives | Concept Indicators Keywords | Source |
|---|---|---|
| Battery degradation | *lifetime, life-time, degradation, aging, cell* | [6,46,47] |
| Cost advantage | *market, markets, day-ahead, cost, aggregator, valley, price, prices, auction* | [43–45,55] |
| Social aspects | *social, fairness, community* | [43,53,56] |
| Integration of renewable energy sources | *pv, wind, renewable, RES, pollution, emission, emissions, solar, environment* | [3,8,39] |
| Congestion management | *load curve, flattening, peak demand, duck-curve, peak, congestion, bus, feeder* | [42] |
| Ancillary services | *frequency, voltage, power quality, loss, losses, current, flow, reactive, security* | [8,38,40,49] |

Table 5 shows the resulting data structure after the screening of the literature. As an example, it holds the three most cited articles found in the 924 results. The first paper, Deilami et al. [48], proposes a real-time coordination mechanism to control multiple BEVs charging to minimize generation costs and grid losses. Besides ancillary services (i.e., minimization of grid losses), the keyword indicator also recognizes the objective cost minimization (i.e., the generation costs). Meanwhile, Sortomme et al. [57] only classifies as ancillary services. Indeed, the paper describes the coordination of electric vehicles to minimize distribution system losses. Gan et al. [58] use a decentralized algorithm to coordinate charging between a utility company and car users. The algorithm shifts the charging loads to fill the valleys to avoid congestion.

Many papers assume the existence of an integrated system operator who is concerned with both a *grid* and a *market-centered* perspective. Consequently, such papers often consider more than one objective in smart charging (i.e., generation cost reduction and congestion management). Out of 511 papers which abstracts include keywords for energy costs, 423 also show a keyword from another objective. In doing so, financial optimization is often the main objective (e.g., if loading flexibility is used to minimize the costs of generation and line losses). Aligning consumption with the generation from RES can also reduce costs. Therefore, a combination of energy costs and RES integration also emerge quite frequently (221 times). Congestion management and other auxiliary services are mentioned at a similar frequency. Fewer papers address keywords describing battery degradation (<20%) or social aspects (<7%) in their abstracts.

As illustrated in Figure 3, there is a rising interest in all topics over time. Even when the first paper mentioning scheduling charging of BEVs found in the review origins from 1980 in Schallenberg [59] a broader discussion of smart charging starts around 2008. In these years, the first serial production BEVs with lithium-ion battery systems (e.g., Tesla Roadster) came to market and spiked a new interest in BEVs and smart charging. However, there is no clear trend for different objectives over time.

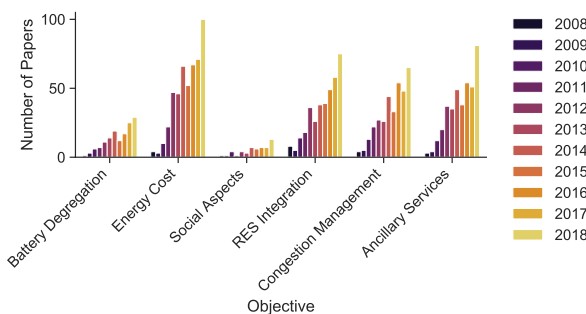

**Figure 3.** Occurrence of keywords in the literature from 2008 to 2018.

**Table 5.** Results of the literature review.

| Source | Objective | | | | | |
|---|---|---|---|---|---|---|
| | Congestion Management | Ancillary Services | Battery Degregation | Energy Cost | RES Integration | Social Aspects |
| Deilami et al. [48] | | ● | | ● | | |
| Sortomme et al. [57] | | ● | | | | |
| Gan et al. [58] | ● | | | | | |
| | 369 | 423 | 169 | 511 | 378 | 57 |

*3.4. BEV Drivers Motivation to use Smart Charging Systems*

The slow rise of papers considering fairness and social aspects can be seen as an indicator that the BEV driver's perspective on smart charging has not been considered to the same amount and with the same rigor as the technical aspects. As the flexibility used in smart charging depends on the BEVs drivers' decisions, it is essential to convince them to use such systems. While financial reward might not be sufficient to convince BEV drivers to use smart charging [8], they usually have limited knowledge about the energy system [60] and the benefits of smart charging. Therefore, it remains an open challenge to identify the objectives that can convince BEV drivers to use smart charging.

Will and Schuller [8] provide a review of 12 studies that research the acceptance factors of smart charging. Most of the studies consider the financial (monetary incentives) and socio-environmental (RES integration) dimension as positive factors that increase acceptance of smart charging. Three out of 12 studies postulate technical aspects (contribution to grid stability) as a motivation factor. For instance, the same paper [8] present a survey with 237 BEV users and find a positive influence of RES integration and grid stability on the acceptance of smart charging systems. Meanwhile, survey results do not show a positive influence of monetary incentives.

In contrast, in an interview survey with BEV drivers that used smart charging systems in a field trial, Schmalfuß et al. [61] find that financial benefits are an essential motivational driver for the usage of such systems. The participants further name RES integration, contribution to grid stability, awareness of energy consumption, and satisfaction from gamification. To gain more recent insights, we perform a forward search for papers citing these two sources [8,61].

Sovacool et al. [6] provide a review that presents a socio-technical approach for V2G charging. The authors distinguish three types of user intervention in smart charging: Time-of-use pricing, where the user receives price signals and actively decides when to charge her BEV. Revenue sharing, whereby users enter their flexibility in for the charging process and receive financial compensation in return. Last, a voluntary shift in charging based on education and non-material motives. The socio-technical dimensions for smart charging objectives are technical, financial, and socio-ecological pictured in Figure 2. The authors describe BEV users' perception of all the above factors as the behavioral dimension of smart charging and conclude that environmental benefits alone will not succeed in convincing BEV drivers to use smart charging.

A review provided by Franke et al. [62] analyzes the BEVs drivers interaction with BEVs and their charging behavior in particular. They find that users have individual time-stable differences in the way the drivers charge their cars. They further analyze that interaction with smart charging systems is costly for the user (with reduced mobility flexibility and increased planning effort) and therefore suggest a user-centered design of smart charging systems. The authors stress two points of a user-centered design: (i) Smart charging systems must provide user guidance and assistance in minimizing effort for the user and (ii) they need to consider the users' objectives in the charging process.

One of the few surveys on the end users' perspective is a discreet-choice experiment by Geske and Schumann [63] among 611 (conventional) vehicle users, including 14 BEV drivers. The acceptance of uncontrolled and smart charging is higher than for V2G concepts. Financial and socio-ecological aspects are the main motivating factors for drivers to use such concepts. As drivers lack understanding and interest in the technical details of the electricity system [60] they are not motivated by technical aspects (e.g., avoidance of grid congestion and reserve power plants).

Tamis et al. [64] describe insights from 11 smart charging projects focusing on smart charging in households in the Netherlands. The objectives of these projects are mainly financial and socio-economical. Five out them focus on two objectives at the same time (e.g., lowering energy cost while using more local renewables). Two of the projects explicitly focus on the community aspect in the socio-economical dimension, and almost all of the project try to benefit more than one stakeholder (e.g., end-user, DSO, municipality, or aggregator) at the same time.

Considering the objectives of BEV fleet operators Ensslen et al. [7] propose a new tariff design for smart charging based on a survey with fleet operators and BEV drivers. Both BEV drivers and

fleet operators focus on the importance of mobility needs for BEV drivers, preferring a minimum level for SoC of 100 km for emergencies. In an earlier study [65], the same authors focus on fleet owners' willingness to pay for smart charging services. Both the guarantee of a minimum level for SoC and the use of higher shares of renewable energies to minimize $CO_2$ emissions have a positive effect on their willingness to pay for smart charging services.

Based on literature, the design of smart charging systems should focus on fulfilling mobility needs with high convenience and security. Financial discounts and the integration of renewable energies are the main motivators to use such systems.

### 3.5. Fit between Operator and User Objectives

The reviews and surveys identified in the literature indicate that financial benefits and integration of renewable energies are the main drivers of acceptance of smart charging systems. The focus on these two objectives omits an evaluation of the motivational power of other smart charging objectives (i.e., battery degradation, social aspects, congestion management, and ancillary services).

Next, we connect these objectives to the perspective of the BEV users to find whether they could also increase acceptance of smart charging if communicated understandably and attractively. Table 6 maps the identified objective functions from *RQ1* with arguments that could convince users to use smart charging systems from *RQ2*.

The arguments are based on the results of the studies collected in the previous section (mainly *'vehicle ∧ charging ∧ incentives'*) or from related research areas such as energy conservation in households. We discuss the objectives in the order of Table 6.

Battery degradation is a big concern to many BEV users [61]. Therefore, smart charging systems capable of reducing battery degradation (as proposed in Schoch [46]) could incentivize drivers to use smart charging.

Many studies [4,7,55,61] find that cost benefits motivate users to use smart charging. To align the operators' financial interests with those of the BEV drivers, Sovacool et al. [6] propose revenue sharing concepts to make the user more flexibility in charging.

So far, most papers omit aspects of fairness and community building in the design of smart charging systems. Meanwhile, research on energy consumption in households has shown that normative information and feedback on neighbors' electricity consumption can reduce electricity consumption by households. Similarly, aspects and the idea of sharing the power grid within a community could also provide an incentive to charge more flexibly [9].

The integration of a higher percentage of renewable energies is an essential driver for users to accept smart charging. Is has several positive aspects and can be framed towards the user from different angles. First, transparent information about the share of renewable energies in the energy supply mix can motivate and influence users. Therefore, Germany specifies electricity suppliers to print its generation mix on the customers' electricity bill. Second, the displacement of conventional power plants reduces emissions of air pollutants. Studies on energy savings show that households consume less energy if they receive information that this behavior reduces air pollutant emissions, thereby preventing respiratory diseases [66]. Third, in addition to air pollutants, carbon dioxide emissions can be avoided by load shifting, which could motivate users to use less energy or be more flexible in consumption, i.e., accept a longer deadline for the charging process.

While the integration of renewable energy has many advantages that are easily understood by BEV drivers, it seems much harder to communicate the benefits of congestion management and grid operation. Similarly, Will and Schuller [8] group concerns about grid stability and find a positive influence on the acceptance of smart charging. We assume that the end-user does not differentiate between voltage quality, frequency, thermal overload, and other grid stability problems. Subsequently, we summarize congestion management and provision of auxiliary services with the term grid impact.

In summary, the objectives of smart charging studied in the literature overlap well with the incentives that could convince BEV drivers to use smart charging systems. We illustrate this, in Table 6, which presents the mapping of promising incentives and smart charging objectives.

**Table 6.** Mapping of smart charging objectives with possible incentives.

| Objective | Incentive | Source |
|---|---|---|
| Battery degradation | Battery degradation | [46,61] |
| Cost advantage | Cost advantage | [4,7,55,61] |
| Social aspects | Social aspects | [56] |
| | Integration of RES | [8] |
| Integration of RES | Environmental protection | [56] |
| | Health impact | [66] |
| | Climate impact | [3,67] |
| Congestion management and ancillary services | Grid impact | [8] |

## 4. Expert Survey

The literature review shows that for charging system operators, energy costs, integration of renewable energy resources, and auxiliary services are the main objectives of smart charging. At the same time, there are only a few studies that research which factors will convince BEV users to use smart charging systems. Although the incentive factors for the BEV users seem to agree with the objective functions of the operators, there is no clear picture of what the most convincing motivational factors are. Studies even contradict each other, for example, Schmalfuß et al. [61] find a positive effect of financial incentives while Will and Schuller [8] do not. Therefore, we survey domain experts to examine their assessment of the potentials of incentive factors. Therefore, we validate the findings from the literature by comparing them to the opinions of experts.

### 4.1. Research Design

To identify the most promising incentive factors and objectives, we first grouped the different arguments for smart charging into eight groups based on the results of the literature review (see Table 6). For each group, we derive three to five one-sentence statements proclaiming the benefits of smart charging. After a revision round discussing the statements in a round of three scientists convened with behavioral economics and electric mobility, we resulted in 31 statements. Getting the agreement of the experts to assess statements is a standardized procedure, which is also used in the Delphi method to reach consensus between expert opinions [68].

We then design an online survey for the evaluation of each statement. In the survey, the domain experts rate each statement regarding its technical accuracy and expected persuasiveness towards end users.

We distributed the survey within a German state-funded research project (https://www.csells. net/) and other channels to professionals in the domain of electric mobility. The survey was online from 30.07.2018 to 6.8.2018. The 16 completed survey included researchers in electric mobility (10), OEMs (1), grid operators (3), and consultants for electric mobility providers and the energy sector (2). As in incentive for each completed survey, we donated 5 Euros to a non-governmental organization (https://www.akcjamiasto.org) concerned with sustainable mobility in Wrocław, Poland.

In the following, we report the English translations of the responses. The evaluation of the statements are rated on a five-level Likert scale from disagreement *strongly disagree (stimmt nicht)* to *strongly agree (stimmt völlig)* to this statement being technically correct or persuasive (compare Döring and Bortz [69]). We operationalize perceived technical accuracy by agreement on *In my opinion, this statement is technically correct. (Diese Aussage scheint mir fachlich korrekt).* The perceived persuasiveness towards end users by agreement on *In my opinion, this statement can convince users (Diese Aussage scheint mir Nutzer überzeugen zu können).*

Example statements for the distinct incentive factors are provided in Table 7. At the end of the survey, participants rated their ability to evaluate the statements correctly and stated their domain background.

**Table 7.** Translation of examples for incentive statements used in the survey.

| Incentive | Example Statement |
|---|---|
| Battery degradation | *Flexible charging can help protect the battery.* |
| Cost advantage | *Flexible charging allows the user to benefit from lower electricity prices.* |
| Social aspects | *The power grid is shared with other users and benefits from the fact that they are flexible when charging BEVs.* |
| Integration of RES | *If users provide charging flexibility, the BEV can be charged with more solar and wind power.* |
| Environmental protection | *Flexible charging allows more electricity from renewable energy sources to be used, thus protecting the environment.* |
| Health impact | *Charging flexibility can avoid conventional generation and thus save harmful emissions.* |
| Climate impact | *Additional temporal flexibility can make a positive contribution towards mitigating climate change.* |
| Grid impact | *Flexible charging contributes positively to grid stability.* |

## 4.2. Results

In Table 8, the incentive factors are listed based on the highest ranking in the two categories. It becomes evident that the two categories are ranked similarly—i.e., what is viewed as correct is also viewed as persuasive with only minor differences. The incentive factors 'Integration of RES', 'Cost advantage' and 'Environmental protection' rate with the highest persuasiveness (ranging from 4.4 to 3.9 out of the maximum of five points). These groups, along with 'Grid impact' are also the top-rated in their accuracy (ranging from 4.4 to 4.2).

**Table 8.** Ranking of groups based on accuracy and persuasiveness rating.

| Group Ranking | Mean Accuracy | Group Ranking | Mean Persuasiveness |
|---|---|---|---|
| Grid impact | 4.4 | Cost advantage | 4.4 |
| Integration of RES | 4.4 | Integration of RES | 4.1 |
| Cost advantage | 4.3 | Environmental protection | 3.9 |
| Environmental protection | 4.2 | Climate impact | 3.6 |
| Climate impact | 3.8 | Grid impact | 3.5 |
| Health impact | 3.6 | Social aspects | 3.3 |
| Social aspects | 3.4 | Health impact | 3.1 |
| Battery conservation | 2.9 | Battery conservation | 2.9 |

The domain experts were very confident in their evaluation: 13 out of 16 agreed or agreed strongly to the statement that they could correctly assess the technical accuracy of the statements, only 3 out of 16 said they partly agreed.

## 4.3. Discussion

Figure 4 shows a scatter plot of all statements in the two dimensions. The plot shows that the statements belonging to one incentive always cluster tightly. The closeness indicates that the pre-selection generated similar statements that belong to the same incentive.

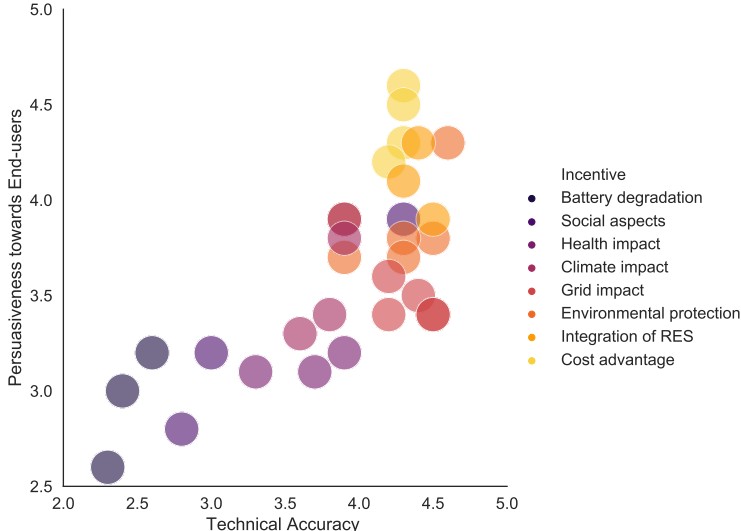

**Figure 4.** Statements evaluated on their technical accuracy (*x*-axis) and persuasiveness towards end users (*y*-axis).

However, there is one exception visible in Figure 4: One statement regarding the social aspects of smart charging rates much higher than the other two. The statement rated higher reads: *The power grid is shared with other users and benefits from the fact that they are flexible when charging BEVs.* It combines social aspects (i.e., use of a common good) with a positive impact on the grid. On the contrary, the other statements are normative messages without any mentioning of other objectives (*BEV users agree that charging should be flexible.* and *Others of the charging station usually allow smart charging.*). While the technical experts rate these normative statements low in persuasiveness, studies by Schultz et al. [70] show that normative framings can persuade end users to behave more environmentally friendly.

In summary, the results show that experts consider those smart charging schemes as most persuasive, which they consider technically correct. Moreover, cost savings and integration of renewable energies are rated highest on both scales. This result corresponds to the high number of papers with these objectives found in the literature. The integration of renewable energy resources has indirect befits (i.e., climate, heath, air pollution) that score lower than statements regarding the integration of renewable energy resources itself.

A positive impact of smart charging on battery degradation is rated as rather low and not very convincing. In one expert opinion, the OEMs already optimized charging to ensure long battery life and therefore no further improvements could be achieved by smart charging.

As the sample of experts is rather small and biased towards research, this might explain why the experts' opinion is in line with the objectives found in the literature. However, the industry experts to no deviate in their responses. As there was a little deviation in the experts' opinions, a method for generating consensus (e.g., Delphi Method) is not needed. Based on the congruent findings in all three research questions, we conclude that cost advantages, RES integration, and grid impact matter most, to both charging station operators and end users.

## 5. Conclusions and Outlook

In this paper, we conduct a literature review to describe the technical potentials of smart charging and identify the objectives for charging station operators and incentives for BEV users to use smart charging systems. The technical preconditions allow for the use of the existing flexibility in BEV charging. At the same time, different potential charging system operators intend to use this flexibility for different objectives.

The objectives depend on the perspective of the charging station operator. Distribution, transmission, or integrated system operators often have a grid-centered perspective on focus on the

provision of auxiliary services and congestion management. Moreover, integrated system operators and aggregators have a market-centered perspective with the main focus on energy costs and partition on flexibility markets. Last, a charging system operator with a local focus might increase the share of renewable energies, consider social aspects, or prevent battery degradation.

Out of these objectives, literature names cost advantage and integration of renewable energy resources as the most significant incentives for BEV users to use smart charging systems. This ranking fits the domain experts' assessment that smart charging can contribute primarily to cost reduction and the integration of renewable energy resources. At the same time, they assess cost reduction and the integration of renewable energy resources as most convincing towards end users. However, this assessment contradicts the findings of Will and Schuller [8], where BEV users stated that financial incentives were not relevant for early adopters of BEVs. Although the avoidance of grid congestion is a relevant area of application, experts doubt that this objective can convince users to use smart charging.

Cost reduction and integration of renewable energies are shared objectives of BEV drivers and charging system operators. These objectives can directly incentivize BEV drivers to use smart charging. However, there are different ways to communicate the benefits of the objectives towards the BEV user. For instance, integration of renewable energy resources can be framed as having health or climate benefits. Field experiments and surveys with BEV users could help to understand which framing is the best to inform and convince the users [71]. Therefore, the design of the user-friendly smart charging systems should not only consider the operator's optimization objectives but also how these targets are communicated to the BEV drivers.

**Author Contributions:** Conceptualization, J.H. and D.J.; Methodology, J.H. and D.J.; Software, E.S.; Validation, J.H., D.J. and E.S.; Formal analysis, E.S. and J.H.; Investigation, E.S.; Resources, Christof Weinhardt; Data curation, J.H. and E.S.; Writing–original draft preparation, J.H.; Writing–review and editing, J.H., E.S., D.J. and C.W.; Visualization, J.H.; Supervision, C.W.; Project administration, C.W. and J.H.; Funding acquisition, C.W.

**Funding:** This research was funded by the Germany Federal Ministry for Economic Affairs and Energy grant number 03SIN135.

**Acknowledgments:** The authors want to thank the Germany Federal Ministry for Economic Affairs and Energy for the funding and support. This research was partly financed by the Smart Energy Showcases-Digital Agenda for the Energy Transition (SINTEG) program.

**Conflicts of Interest:** The authors declare no conflict of interest. Dominik Jung is from Porsche AG, the company had no role in the design of the study; in the collection, analyses, or interpretation of data; in the writing of the manuscript, and in the decision to publish the results.

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
