# Peer review of "Quo Vadis Smart Charging? A Literature Review and Expert Survey on Technical Potentials and User Acceptance of Smart Charging Systems"

_wevj, doi:10.3390/wevj10040085_

Round 1

Reviewer 1 Report

On substance, a larger survey sample would enhance the value of the conclusions to RQ3. On editing, please note that the table on p.3 contains an error: the Chevy Bolt has a 60kWh battery (not 6)    

Author Response

Response to Reviewer 1 Comments

Point 1.1: On substance, a larger survey sample would enhance the value of the conclusions to RQ3

Response 1.2: Please see Response 2.1.

Point 1.2: On editing, please note that the table on p.3 contains an error: the Chevy Bolt has a 60kWh battery (not 6)

Response 1.2: This is correct. We shifted the decimal point.

Reviewer 2 Report

Dear authors, The review and survey work are interesting and useful, and it is a hard work to do such a survey work. Generally speaking, survey is a probability statistics issue. The rationality of samples (the number, the type of samples, etc..) is very important. In order to know the answers of RQ1,2,3, may be you need to do a lot of survey work. I don’t know how much sample data is persuasive and reasonable, the samples 16 survey (in line 493) or 16 experts (in abstract) can support your conclusion or not? and why end-users are not considered directly in your survey work. you need to explain. best regards

Author Response

Response to Reviewer 2 Comments

Point 2.1: The rationality of samples (the number, the type of samples, etc..) is very important. In order to know the answers of RQ1,2,3, may be you need to do a lot of survey work. I don’t know how much sample data is persuasive and reasonable, the samples 16 survey (in line 493) or 16 experts (in abstract) can support your conclusion or not?

Response 2.1: We address the research question one by one:

In our opinion, RQ 1 (What are the objectives of charging system operators present in academic literature?) does not require a statistical analysis, which would be uncommon as it is answered with a literature review. As a sample we use three literature databases that are relevant in the field.

The second research question (“What incentives motivate BEV drivers to use smart charging systems?”) was probably a little bold. In fact, reviewing the related work can (at this point in time) not provide a full answer and would require a survey with end-user (see Point 2.2). Therefore, we changed the research question to “What are promising objectives to motivate BEV drivers to use smart charging systems?”.

We acknowledge that both, the selection and the number of experts are not extensive enough to answer RQ3 on its own account. This would require another method (e.g. Delphi) or a much larger sample size. However, at this point, we use the survey to verify our findings for the first two research questions. The key point is: Literature find renewable integration and cost reduction as the most important topic for both, end-users and operators When we asked possible operators and developers, they said the same (added line 482). We added a paragraph in the discussion (4.3 – line 542) to make this clearer and acknowledge the limitations.

Point 2.2 and why end-users are not considered directly in your survey work. you need to explain

Response 2.2: Thank you for raising this question. The point-of this survey is to narrow down the search space of smart charging objectives, that work well with end-users. In particular, we want to find out, whether there is consensus or divergence between different experts in the field.

However, we think that the first important feature of a successful smart charging system is that the operator implements a rewarding objective function. Out of this subset, the operator has to find an objective function that also works with end-users (e.g. conducting surveys or field experiments). However, we see do not consider this in the scope of this paper. In fact, we used the expert survey to derive objectives for a survey with 164 BEV-drivers:

Huber, J.; Jung, D.; Schaule, E.; Weinhardt, C.  Goal framing in smart charging - Increasing BEV users’ charging flexibility with digital nudges.  Proceedings of the 27th European Conference on Information Systems (ECIS), Stockholm and Uppsala, Sweden, June 8-14, 2019, 2019

Reviewer 3 Report

This paper analyses the user acceptance of smart charing systems based on literature review and expert survey. The topic is very timely and methodology is unique. Also, the logic and conclusion are scientifically sound. The reviewer recommend this manuscript for the publication of WEVJ. 

Author Response

Response to Reviewer 3 Comments

No issues raised.

Round 2

Reviewer 2 Report

No recommendation.